



# The Capability of high spatial-temporal remote sensing imagery for monitoring surface morphology of lake ice in Chagan Lake of Northeast China

Qian Yang[1, 2], Xiaoguang Shi[1], Weibang Li[1], Kaishan Song[2], Zhijun Li[3], Xiaohua Hao[4], Fei Xie[3], Nan Lin[1], Zhidan Wen[2], Chong Fang[2] and Ge Liu[2, *].

[1]Northeast Institute of Geography and Agroecology, Chinese Academy of Sciences, Remote Sensing and Geographic Information Research Center, Changchun, 130102, China
[2]Jilin Jianzhu University, School of Geomatics and Prospecting Engineering, Changchun, 130118, China
[3]Dalian University of Technology, State Key Laboratory of Coastal and Offshore Engineering, Dalian, 116024, China
[4]Northwest Institute of Eco-Environment and Resources, Chinese Academy of Sciences, Lanzhou, 730000, China

*Correspondence to*: Ge Liu (liuge@iga.ac.cn)

**Abstract.** The surface morphology of lake ice undergoes remarkable changes under the combined influence of thermal and mechanical forces, which has been rarely observed by remote sensing. A large-scale linear structure has repeatedly appeared on satellite images of Chagan Lake in recent years. We prosed a method to extract linear structure on the lake ice surface. We applied it to high spatial-temporal images merged by the Landsat and GOCI images using an enhanced spatial and temporal adaptive reflectance fusion model (ESTARFM). We monitored the changes in surface morphology in Chagan Lake from November 2018 to March 2019, which were further verified as ice ridges during the field investigation. The average length of the ice ridges during the completely frozen period was 21141.57 ± 68.36 m. The average azimuth angle was 335.48° ± 0.23°, perpendicular to the wind domain. Besides, we discovered spherical ice balls along the southwestern coast. The deformation of surface morphology is closed related to wind direction, snowfall, and air temperature.

**Keywords:** remote sensing, lake ice, GOCI, surface morphology, wind, ESTARFM

## 1 Introduction

The lake ice is one of the Essential Climate Variables in the cryosphere (Bojinski et al., 2014), which is closely associated with the lake environment, ecological regulation, public transportation, and ice activities safety (Hampton et al., 2017; Magnuson et al., 2000; Leppäranta, 2015; Brown and Duguay, 2010; Arp et al., 2020). Shorter ice cover duration and thinner ice thickness have been a common trend throughout the world (Ipcc, 2021; Srocc, 2019; Murfitt and Duguay, 2021). Recent work using remote sensing mainly focused on lake ice phenology (Weber et al., 2016; Zhang et al., 2021; Xie et al., 2020; Murfitt and Duguay, 2020; Du et al., 2017), lake ice classification (Tom et al., 2020; Hoekstra et al., 2020), ice thickness (Murfitt et al., 2018b; Kang et al., 2014; Gogineni and Yan, 2015) and ice albedo (Li et al., 2018; Lang et al., 2018). However, scarce work studies the surface morphology of lake ice, i.e., ice ridges and lake ice fracture. The surface morphology is



controlled by dynamic processes of lake ice, which have attracted widespread attention from academia and the public. This paper monitored the surface morphology of Chagan Lake, Northeast China, combining high spatial-temporal remote sensing data and field investigation, and explored the potential influence of climate factors.

Satellite remote sensing has the advantages of macroscopic, multi-source, and wide-range and has been successfully applied in global remote sensing monitoring of lake ice (Murfitt and Duguay, 2021; Doernhoefer and Oppelt, 2016; Du et al., 2019). Visible light and multispectral data were first used for lake ice monitoring based on the spectra difference between ice and water, but the best period for monitoring lake ice changes is easily missed due to the influence of clouds, fog, and light (Howell et al., 2009; Cai et al., 2019; Yang et al., 2019; Qi et al., 2020). Active microwave data identify the presence of ice through the differences in the backscatter of water and ice, while passive microwave through differences in bright temperature (Cai et al., 2017). Microwave remote sensing can penetrate the dry snow on the lake ice surface to observe the internal structure and stratification of lake ice (Jones et al., 2013), from the early stage of qualitative differentiation between ground ice and floating ice to quantitative inversion of lake ice phenology and thickness (Ke et al., 2013; Jeffries et al., 2013; Howell et al., 2009; Kang et al., 2014). Although the temporal resolution of the active microwave remote sensing data has been improved from 30 days (ERS) to daily return visits (Radasat-2), the cost is too expensive and more suitable for case studies for small or medium lakes (Murfitt et al., 2018a; Geldsetzer et al., 2010). With high temporal resolution and time series, passive remote sensing data can detect ice cover under all weather conditions and is limited by low spatial resolution and significant mixed image effects. Thus, passive remote sensing is more suitable for large-scale lake ice monitoring (Du et al., 2017; Qiu et al., 2018). In summary, multi-sensor, multi-temporal and multi-spatial resolution remote sensing image data have been successfully applied to monitoring the lake ice. Still, single-sensor remote sensing data cannot simultaneously balance clear and accurate remote sensing monitoring with high frequency.

Lake ice's growth and decay process change very fast, requiring high temporal resolution to capture surface morphology. The satellite sensor with moderate spatial resolution, such as VIIRS and MODIS, can monitor the temporal changes of lake ice daily but fail to reflect spatial details of surface morphology. The satellite sensors with medium spatial resolution, such as Landsat and Sentinel, could provide fine texture, but frequent images are not available to capture the fast changes. The fusion methods consist of unmixing, weight function, and dictionary-pair learning methods (Sisheber et al., 2022; Zhu et al., 2016). The most common weight function method includes the Spatial and Temporal Adaptive Reflectance Fusion Model (STARFM) (Feng et al., 2006), spatial and temporal adaptive algorithm (STAARCH) (Hilker et al., 2009), enhanced version Enhanced Spatial and Temporal Adaptive Reflectance Fusion Model (ESTARFM) (Zhu et al., 2010), flexible spatiotemporal data fusion (FSDAF) model (Zhu et al., 2016), and so on. Previous studies have proved that these spatial-temporal fusion methods could improve the monitoring abilities of remote sensing for specific applications but fail to monitor the abrupt changes in landscapes and spectral differences. All these models are derived by pairs of coarse and fine resolution and coarse resolution images, e.g., one for STARFM, and two for ESTARFM. ESTRAM performs better than STRAFM model in heterogeneous landscapes (Zhu et al., 2010; Knauer et al., 2016; Wang et al., 2021b; Jarihani et al., 2014). STAARCH could require two pairs of bases strictly when monitoring the spatial changes, one before and one after the changes. FSDAF is more robust than the other three methods



but has limited detection for the tiny changes (Zhu et al., 2016), which make it difficult to monitor the surface mopholgoy changes. Therefore, we generated fusion images with high spatial-temporal resoluiton based on ESTARFM model for further exploration.

The evolution of a lake ice season is primarily a thermodynamic process under the influence of wind forcing, water level variations, and thermal forces. The thermal forces enable the surface to melt and freeze when the day and night alternate and

the mechanical strength of wind and flow make the ice bulk move and collide, and then water course, ice ridges, and ice fractures appear, develop and disappear. Therefore the surface morphology of lake ice exhibits spatial-temporal difference periodically, which differs significantly from the flat and smooth surface of lake ice. The horizon and linear structure of lake ice was monitored by optical satellites for large lakes in Europe, which were explained by ice displacement (Leppäranta, 2015). The high-resolution satellite-airborne SAR images have been used to monitor sea ice surface deformation, such as ice ridges

(Dierking, 2010). Moreover, airborne aerial platforms have effectively complemented remote sensing data sources for monitoring changes in lake ice morphology, such as UAVs (Li et al., 2020), airborne radar (Jeffries et al., 2013), and ground-penetrating radar (Gusmeroli and Grosse, 2012). The spatial distribution of lake ice surface morphology presents complex, variable and discontinuous characteristics, highlighting linear features in remote sensing images. At present, few studies study the changes in surface morphology of lake ice and the impact factors, so it is meaningful to develop a quantitative method to

describe the lake ice surface morphology.

This study proposed the workflow based on ESTARM fusion images to monitor surface morphology in a highly dynamic process. The objectives of our work are to (1) propose a reliable method to extract the surface morphology of lake ice; (2) monitor the changes in surface morphology for Chagan Lake using high spatial-temporal satellite imagery; (3) explore the formation mechanism of surface morphology observed from satellite and field investigation.

**2 Materials**

**2.1 Study area**

The Chagan Lake basin (124°03′-124°34′E, 45°09′-45°30′N) (Figure 1), one of the ten largest lakes in China, plays an essential role in fisheries, agricultural irrigation, and winter recreation in the surrounding areas (Wen et al., 2020). The average and maximum water depths are 2.5 and 4.5 meters (Duan et al., 2007; Song et al., 2011). The lake has a water area of 252.86 km$^2$

and a length of 107.28 km from Landsat OLI 8 on January 10, 2019. The lake area is characterized by semi-arid and sub-humid continental monsoons with air temperature, precipitation, and evaporation of 5.5 ℃, 430 mm, and 1496 mm (Song et al., 2011; Duan et al., 2007). The salinity ranges from 0.31 to 0.78 ‰ (Liu et al., 2020).

The recharge sources mainly consist of precipitation, groundwater, and adjacent irrigation discharge (Liu et al., 2019). The bottom of Chagan Lake is relatively flat, the soil is primarily powdery-sandy with the type of white calcium-alkaline, and

salinized soil farmlands and grassland pastures are widely distributed in the catchment area. The Chagan Lake is a typical seasonal frozen lake, and the existence of ice cover lasts from November to April each cold season, with the maximum ice



thickness ranging from 0.8 to 1.1 meters (Liu et al., 2020; Hao et al., 2021). We conducted two field investigations from December 30 to 31, 2020, and January 2 to 4, 2022. The goal was to verify the linear structure observed from remote sensing images. We measured the ice thickness by an electronic digital calliper with a resolution of 0.01 mm. Besides, we also
measured the water depth by hand-held sonar detector SpeedTech SM-5 with a resolution of 0.1 meters and compared the difference between autumn (September 17, 2021) and winter (January 2 to 4, 2022).

**[Figure 1 is added here]**

## 2.2 Materials

### 2.2.1 GOCI

The Geostationary Ocean Color Imager (GOCI) was the first satellite for detecting ocean colour from a geostationary orbit, which has been widely applied in deriving the optical, biological, and biogeochemical properties (Ryu et al., 2012; Ryu and Ishizaka, 2012). GOCI data have been available since April 2004, covering about $2500 \times 2500$ km$^2$ around the Korean Peninsula. Its six visible bands range from 412 to 443 and 540 to 555, 660 to 680 nm and two near-infrared bands range from 745 to 865 nm. The GOCI provides eight hourly observations from local times from 8:00 to 15:00 with a spatial resolution of
500 m. The best advantage of COCI is the frequent temporal resolution with eight images daily and could offer details for freeze-up and break-up processes. 96 GOCI images during the cold season of 2018-2019 were used in this paper. The atmospheric correction has been carried out by GOCI data processing software (GDPS).

**[Table 1 is added here]**

### 2.2.2 Landsat

The Landsat 8 satellite was launched on February 11, 2013, carrying the Land Imager (OLI) and the Thermal Infrared Sensor (TIRS), which have been widely used for lake and river ice monitoring (Wang et al., 2021a; Yang et al., 2020), with the OLI having nine bands. It has a spatial resolution of 30 m for Bands 1-7 and Band 9 and a spatial resolution of 15 m for Band 8. It has a temporal resolution of 16 days. Four Landsat images during the cold season of 2018-2019 were prepared for data fusion with GOCI. The capture date is December 08, January 25, February 26, and March 14, respectively. The path and row are 119
and 29. We downloaded the Landsat calibrated surface reflectance Tier 1 collection for Landsat OLI ('LANDSAT/LC08/C01/T1_SR') from Google Earth Engine (GEE) for further work. More details of band information of Landsat and GOCI can be found in Figure S1.

### 2.2.3 Auxiliary data

The cold season is defined from October of the current year to April of next year according to the local climate. The surface
temperature of the lake was extracted from MOD11A1 products for each cold season from 2000 to 2021, following the processing procedure of our previous work (Song et al., 2016; Hao et al., 2021). Daily air temperature, precipitation, wind direction, and wind speed of Qian'an station (ID: 50948) were utilized to explain the influence of climate on lake ice from





2010 to 2021. The longitude and latitude of Qian'an are 124.011° E and 44.998° N, with an elevation of 146.3 meters. The climate records are used to explain the lake ice development. A 16-degree angle with an interval of 25˚ was used to indicate
the direction of the wind.

## 3 Methods

### 3.1 The framework of methodology

Figure 2 presents the flow chart of our work herein. We pre-processed Landsat and GOCI and prepared the reflectance images of Landsat band 2 and GOCI band 3. Then, we merged GOCI and Landsat using the ESTARFM model and generated new
fusion data with a spatial and temporal resolution of 30 meters and 1 hour.  After that, we identified the geographic location of the linear structure on the lake ice surface and extracted the morphological parameters, including length and angle.

**[Figure 2 is added here]**

### 3.2 The estarfm fusion

The enhanced spatial and temporal adaptive reflectance fusion model (ESTARFM) was proposed by Zhu et al. (2010) based
on the STARFM model, which used two pairs of Landsat and GOCI images to generate the spatial-temporal fusion data. Firstly, the coarse GOCI data was projected and resampled to a fine Landsat image at two known times $T_m$ and $T_n$. Secondly, similar neighbourhood pixels were searched by a moving window by setting spectral differences. Thirdly, the normalized weight of each similar pixel was calculated considering the spatial, spectral, and temporal differences. Then, the coarse GOCI values were transferred to fine Landsat data using the pixel-based conversion coefficients from linear regression. Finally, the fine
fusion data at the predicted time ($T_p$) are calculated by the coarse GOCI data at the same time, which are expressed as follows (Liu et al., 2021; Bai et al., 2017; Zhu et al., 2010):

$$L_b\left(x_{w/2}, y_{w/2}, T_p\right) = L_b\left(x_{w/2}, y_{w/2}, T_k\right) +$$

$$\sum_{i=1}^{n} W_i \times v_i \times \left(G_b\left(x_i, y_i, T_p\right) - G_b\left(x_i, y_i, T_k\right)\right) \ \ (k = \ m, \ n) \tag{1}$$

Where $L_b$ and $G_b$ represent the reflectance of MODIS and GOCI images in band $b$, $w$ stands for the moving window size, and the corresponding center is ($x_{w/2}$, $y_{w/2}$). $W_i$ is the weight of a similar pixel contributing to the predicted pixel; $v_i$ is the conversion
coeffects; $T_k$ is the known time, including $T_m$ and $T_n$.

### 3.3 The morphological extraction

The Landsat-GOCI fused images were transformed into binary images. The linear network was extracted by the method of the Canny operator, and then a fractal-based detection was carried out to remove the outer boundaries. The morphological processing was implemented for the inner part of the linear network, including opening, filling, and eroding sequentially. Then,





we divided the largest connected domain of inter parts into two paths, and the shortest path was considered the final length. The angle was determined as the north-south connection line and the north direction along the clockwise direction. We compared the auto-extraction and visual interpierion in our previous work (Hao et al., 2021). Regarding the length and angle of linear structures, auto-extraction performed well with an $R^2$ value of 0.96 and 0.98, respectively.

## 4 Results

### 4.1 The ESTARFM model

Figure 3 displays the comparison of the original images and predicted images on November 22, 2018. The two related pairs of Landsat and GOCI were captured on November 6, 2018, and December 8, 2018. The predicted image kept the texture of the ground objects, which is consistent with the original images. We also enlarged part of the linear structure, showing a good fusion effect. Figure 4 illustrates the scatter plots along the 1:1 line of the actual and estimated reflectance values and the corresponding statical results. The $R^2$ had the value of 0.93, which indicated the predicted image was highly correlated with the actual image. Besides, we compared the reflectance values of original images and predicted images on November 22, 2018. The result showed that the range of estimated and actual images was consistent; the mean of their reflectance values was both $0.10 \pm 0.03$. Therefore, the ESTARFM fusion images had a good performance and provided reliable materials for further exploration.

**[Figure 3 and 4 is added here]**

We predicted the fine images from two pairs of fine Landsat and coarse GOCI data to fill the data gap caused by the low revisit frequency of Landsat. The high $R^2$ between actual and predicted images was 0.935 on November 28, 2018, which proved that the fusion images are consistent with the remote sensing data. Liu (2018) et al. pointed out that when validating the ESTARFM model, a 4-day time lag between fusion and validation results may lead to errors in the analysis of the results. The time lag in our study ranged from 5.5 to 12.5 hours for Landsat and GOCI, and the assumption that the lake ice will not change dramatically in a short period. The performance of Estarfm results was limited by: (1) Only three image pairs were available during the cold seasons; (2) the time lag between the prediction and verification images; (3) the inconsistency between the predicted time and two input pairs (Lu et al., 2019; Liu et al., 2018). The ESARFAM model predicted the reflectance using a linear model and assumes that the reflectance changes steadily during the fusion process (Liu et al., 2018). These limitations need to be considered in our future work.

### 4.2 The changes in surface morphology

We extracted the surface morphology of Chagan Lake from 96 fusion images during the cold season of 2018-2019. Figure 5 display spatial changes of linear structure on Landsat images during the freeze-up and break-up process. Figure S3 and Figure S4 presented the original images from GOCI with a resolution of 500 meters, the fusion images from Landsat and GOCI with



a spatial resolution of 30 meters, the network structure of surface morphology, and the surface morphology during the freeze-up and break-up process, which provide more details the extraction process. The linear structure on image appeared from southeast to northwest, lasting from November 22 to November 30, 2018. The liner structure disappeared from the northwest to southeast, lasting from March 15 to March 24, 2019. The large-scale fracture extending from northwest to southeast that repeatedly appeared on Landsat images since 1986, which has been reported in our previous work (Hao et al., 2021).

**[Figure 5 is added here]**

Figure 9 shows the average daily length of ice ridges from Landsat and GOCI remote sensing data. It can be seen that the ice ridge changes in the winter of 2018-2019 can be divided into three processes, including the growth process, stable process, and recession process. The growth stages lasted from November 22 to November 30, a total of 9 days. The length of the ice ridge changed from 5211.17 meters to 18042.15 meters with an average value of 12680.32 ± 4472.37 meters, extending from

the southeast to the northwest. The average azimuth angle of the ice ridges during the growth process was 334.38° ± 2.08° ranging from 331.54° to 338.17°. The recession lasted from March 15, 2019, to March 24, 2019, a total of 10 days. The length of the ice ridge changed from 19178.18 meters to 5924.03 meters with an average value of 13288.59 ± 4907.89 meters, disappearing from northwest to southeast. The average azimuth angle of the ice ridges during the recession process was 332.90° ± 2.54° ranging from 329.84° to 336.16°. The changing growth and recession process rate were 1425.66 and 1325.42 meters

per day, which suggests the rapid growth is slightly faster than the recession process.

**[Figure 6 is added here]**

## 4.3 The field investigation

The lake ice process is governed by the complex interaction of hydraulics, thermodynamics, and mechanics. The heat loss caused by decreasing air temperature exceeds the heat gained from the surface water in the late autumn and early winter. When

the water temperature falls below the freezing point, the cooling water provides a beneficial condition for ice crystals. Then the volume of lake ice expands and the amount increases, followed by ice formation. The average freeze-up date of Chagan Lake was November 12 during the 12 recent cold seasons from 2010 to 2022, which has been listed in Table S1 (Hao et al., 2021). Considering the traveling safety on ice, we conducted two thorough investigations during the two recent cold seasons, on December 30 to 31, 2020, and January 2 to 4, 2022, respectively. We located ten sampling points along the linear structure

on satellite images and collected field photos of ice ridges and fractures, as shown in Figure 1. We further verified the large-scale fractures on images as ice ridges. More interesting is that we both discovered ice balls along the southwest coast of Chagan Lake in the two years, as shown in Figure 7. The ice ball piled above the ice surface in the winter of 2020, and the ice was so uneven that the cars had difficulties traveling on the ice surface. However, the ice ball was frozen beneath the ice surface, and the surface was relatively smooth.

We also measured the ice thickness and water depth of each sampling point, presented in Figure 7. The ice thickness in the winter of 2022 ranged from 437.55 mm to 668.25 mm, with an average of 582.24 ± 58.14 mm. We divided into three regions: Region 1 was distributed along the ice ridges; Region 2 was distributed along the north-eastern coast; Region 3 covered the





southern part of Chagan Lake. The average values of regions 1, 2, and 3 were 551.58 mm, 547.75 mm, and 645.74 mm. Among the three regions, the ice thickness had the smallest value. The difference between summer 2021 and winter 2021 varied from

0 to 0.2 m with an average value of 0.12 ± 0.05 m. This indicated that winter's water depth is lower than summer's, but the differences were not significant enough to explain what we observed from the satellite. The ice features first formed in the southeast coast's nearshore area, where the water depth was relatively shallower than in other regions in Figure 7. The ice thickness and water depth showed spatial coherence with the surface morphology, especially with the ice thickness.

**[Figure 7 is added here]**

**4.4 The climate condition**

As shown in Figure 8, we analysed the wind rise and daily average and maximum winds from November 1, 2018, to April 15, 2019. The freeze-up and break-up dates of Chagan Lake in the cold season of 2019 are November 17, 2018, and March 22, 2019. The ice ridges appeared on November 22, 2018, 5 days later than the freeze-up date; they disappeared on March 24, 2019, 2 days later than the break-up date. The temporal resolution of MODIS LST products is eight days and can explain the

time difference. We also found that the WSW (247.5°) direction had a relatively high frequency for all three stages. The frequency of WSW was 22.22%, 30%, and 14.23% for the growth, stable, and disappearing processes. The angle between WSW and ice ridges was 87.98°, 86.88°, and 85.4°. The nearly perpendicular relationship was consistent with our previous study (Hao et al., 2021). Moreover, the three quartiles of daily average speed were 3.52 m/s, and those values of the growth process (3.88 m/s) and recession process (3.63 m/s) were more prominent than the three quartiles. The three quartiles of daily

maximum speed were 6.92 m/s, and those of growth process (7.05 m/s) and recession process (6.86 m/s) were close to the three quartiles. Therefore, wind speed and direction played a crucial role in the development of ice ridges.

**[Figure 8 is added here]**

To explain the existence of ice balls, we checked the changes in air temperature, precipitation, wind speed, and wind direction of Lake Chagan around the freeze-up periods from 2010 to 2021, as shown in Figure 9. The freeze-up dates were provided by

MODIS (Table S1), and we calculated the average values covering one week before and after the freeze-up date. The air temperature in the cold season of 2020 and 2021 was not significantly different from other years, but the precipitation was significantly higher than in other years. The values were 20.9 and 12.5 mm in 2020 and 2021, respectively, and further were considered snowfall because of the air temperature below 0℃. In addition, the average wind speed in these two years is 7.2 and 12.3 m/s, which is significantly higher than the other years. More interestingly, the wind direction of the northeast direction

happened during the freeze-up periods for both years, while the southwest wind dominated the other years. We deduced that the northeast wind blew the ice ball to the southwest shore, which created the uncanny geographic landscape we found during the fieldwork. Therefore, the air temperature, snowfall, and wind create favourable conditions for ice ball.

**[Figure 9 is added here]**



## 5 Discussion

The lake ice experienced different phases, including the ice crystal, frazil ice, nails, pancake ice, and ice layer.  The lake ice began to freeze in shallow water areas and expanded to the lake centre until the lake was completely frozen. Lake ice deformed under the combined influence of thermodynamics and hydrodynamics, resulting in lead, cracks, and ice ridges. The lake ice expanded and contracted as the air temperature rose and dropped during the cold season. The temperature difference between night and day results in lake ice's thermal expansion and contracts, which differ significantly within a given lake. Furthermore,

long and narrow cracks then happened and were likely to evolve into ice ridges under the pressure when the lake ice bulked, collided, and piled up. The view range limited traditional field measurement and had difficulty identifying the spatial distribution of ice ridges. In this work, we proposed an algorithm to detect the high-lightened linear structure based on remote sensing images. The recurrent linear structure on a large scale was further verified as ice ridges in the fieldwork. The ice ridges were supposed to happen in the area with thinner ice thickness. During the field investigation, we measured the ice thickness

along the linear structure. We compared it with the other regions (Figure 7), and the difference of ice thickness is not significant to support this, which required a thorough investigation of ice thickness in Chagan Lake by combing the in-situ measurement and remote sensing monitoring.

The development of seasonal lake ice is a dynamic process, and the wind and currents force the shift of ice bulk. During the freeze-up process, the wind and water flow push the ice toward the shore, preventing the ice cover from freezing; during the

break-up process, the wind could break the ice cover and accelerate the melting. The ice ridges underwent three stages during the cold season of 2018, during which the wind direction and speed exhibited remarkable differences in Figure 8.  The ice ridges grew from southeast to northwest with an average direction of 334.38° and decayed from northwest to southeast with an average direction of 332°. The direction of ice ridges is nearly perpendicular to WSW (247.5°), which frequently happened for all three periods, revealing the crucial role of wind in the development of ice ridges. However, more aspects

should be considered. More interestingly, the direction of ice ridges had a similar shape to the southwest shoreline, and the stable shoreline geometry could explain the recurrent ice ridges with a specific direction, which had been reported in previous studies (Leppäranta, 2015).

We also discovered the spherical ice ball in the neighbourhood of the southeast coast, and this rare geographic scenic attracted our interest, which has been reported in Finland, Japan, and so on (Case, 1906; Loewe, 1949; Langlois, 1965; Eisen et al.,

2003; Kawamura et al., 2009).  From Figure 9, the occurrence of ice ball needs to meet the strict requirements of climate conditions together during the frozen process: (1) the air temperature varies up and down the freezing point and can meet the freezing conditions; the temperature cannot be lower than -10 ℃. Otherwise, the ice will directly freeze completely; (2) sufficient snowfall is necessary to provide material sources for the emergence of ice balls, snow and water mixed and frozen together, the final ice balls formed under the continuous washing of wind and waves; (3) wind speed is large enough to push

the rolling spherical ice to southwest coast only if the wind direction was northeast before freezing, and then ice balls aggerated





the nearshore areas (Xie Fei, 2022). During the past decade, only the cold season of 2020 and 2021 could meet all three conditions, and we, fortunately, discovered these ice balls during the field investigation.

## 6 Conclusion

We generated high spatial-temporal remote sensing data from Landsat and GOCI using ESTARFM, and filled the gap for fine monitoring lake ice dynamic in Lake Chagan. We compared the reflectance of the fusion image and the original image on November 22, 2018. The scatter plot was centred around 1:1 with the $R^2$ value of 0.93, which indicated the predicted image was highly correlated with the actual image. Besides, the predicted image kept the ground objects' texture consistent with the original images. Therefore, the ESTRAFM fusion images provided reliable materials for further exploration.

We proposed the automatic extraction and calculated the length and the angle of linear structure on fusion images during the cold season from 2018 to 2019. Based on the satellite stages, the linear structure experienced growth, stability, and decay. The growth lasted nine days, just four days later than the freeze-up date from MODIS; The decay process lasted for 9 and 10 days, just two days earlier than the break-up date. From southeast to northwest, the linear structure was 5211.17 to 18042.15 meters long during growth; from northwest to southwest, it disappeared. The average length of the ice ridges during the fully frozen period was 21141.57 ± 68.36 m. The average azimuth angle was 335.48° ± 0.23°.

We carried out several field investigations and verified the linear structure of the ice ridges, along which we also discovered spherical ice balls. The direction of the ice ridge was nearly perpendicular to the southwest wind direction, which is the dominant wind direction with high frequency. The deformation of surface morphology was related to the meteorological conditions before the freeze-up process, including wind, snowfall, and air temperate. This work demonstrates the capability of monitoring large-scale surface morphology using multi-source remote sensing. We need to simulate the flow field using the
hydrodynamic model and explore the relationship between the hydrodynamic field and the surface morphology of the lake ice in the future.

## Acknowledgment

This research was jointly supported by the National Natural Science Foundation of China (41971325), the 14th Five-Year Plan of Technical and Social Research Project for Jilin Colleges of China (JJKH20210290KJ), and the National Key Research and
Development Program of China (2019YFE0197600).

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



**Tables**

**Table 1 The usage of GOCI images from November 2018 to March 2019.**

| Date | Image number | Date | Image number |
|---|---|---|---|
| 2018.11.21 | 1 | 2019.03.14 | 1 |
| 2018.11.22 | 8 | 2019.03.15 | 6 |
| 2018.11.23 | 7 | 2019.03.16 | 0 |
| 2018.11.24 | 0 | 2019.03.17 | 8 |
| 2018.11.25 | 8 | 2019.03.18 | 8 |
| 2018.11.26 | 0 | 2019.03.19 | 0 |
| 2018.11.27 | 7 | 2019.03.20 | 0 |
| 2018.11.28 | 8 | 2019.03.21 | 0 |
| 2018.11.29 | 8 | 2019.03.22 | 4 |
| 2018.11.30 | 7 | 2019.03.23 | 8 |
| 2019.03.13 | 1 | 2019.03.24 | 7 |



**Figures**

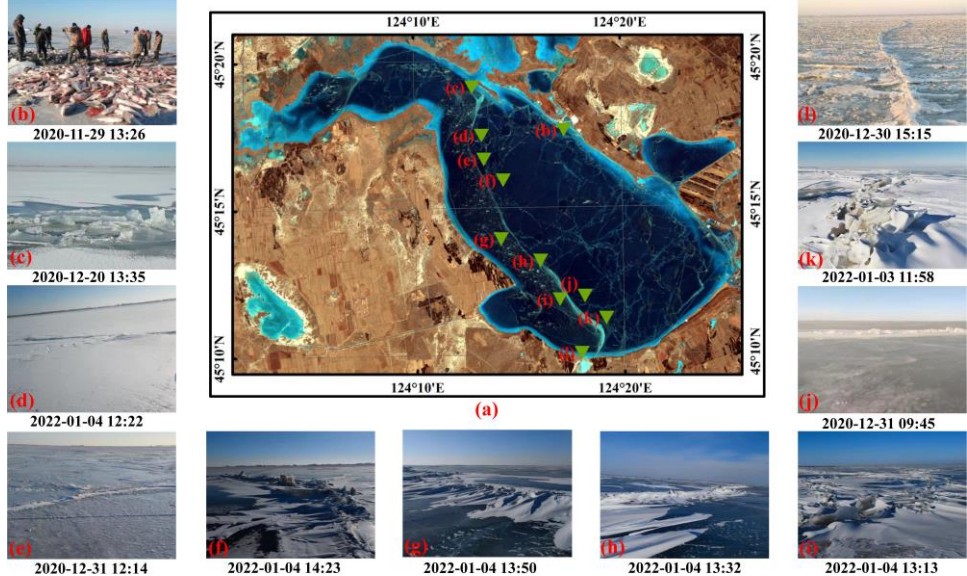

**Figure 1 The spatial distribution of Chagan Lake and field photographs. Figure 1 (a) is provided Landsat 8 OLI on February 10, 2019 with the band composite: R(5) + G(4) + B(3). Figure 1 (b) shows the fishing activities of Chagan. Figure 1(c)-(h) displays the field photographs along the linear structure observed from the satellite images in our two recent investigations.**

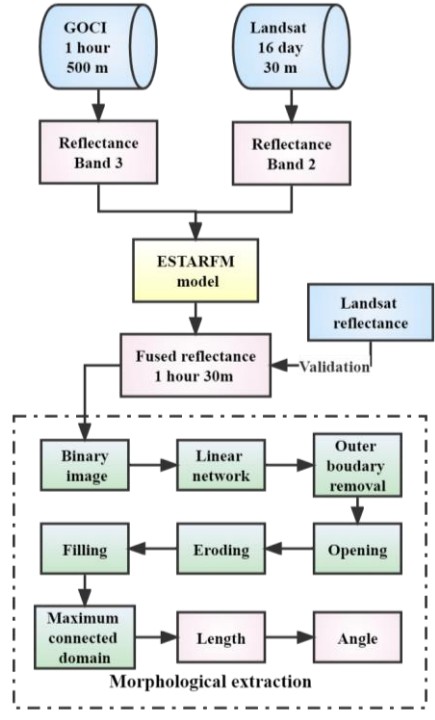

**Figure 2 The flowchart of the methodology of this study.**


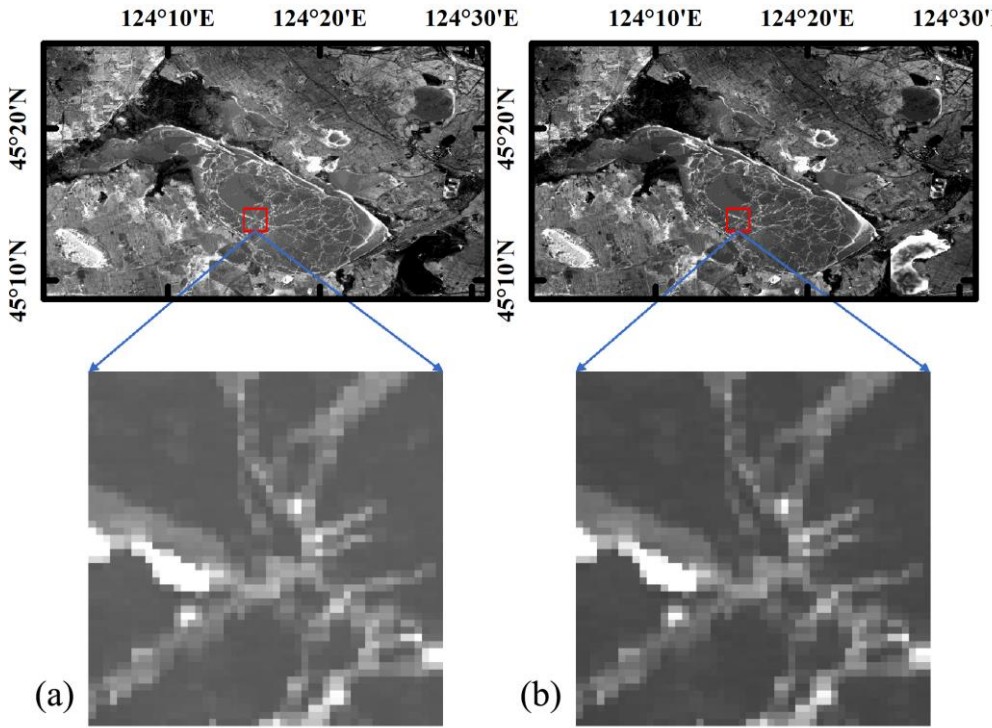


**Figure 3 The image comparison of Landsat 8 OLI (a) and ESTARFM (b) on November 22, 2018.**

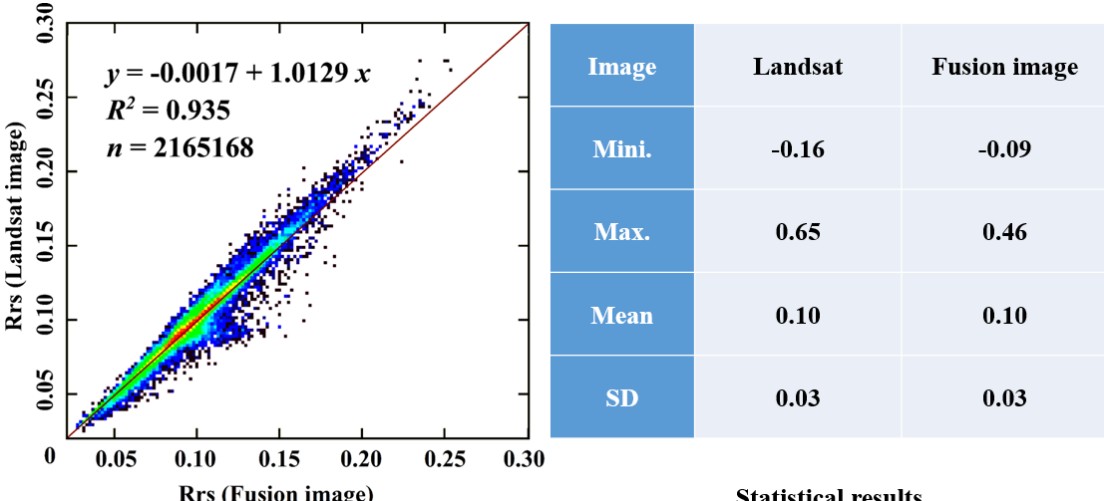

| Image | Landsat | Fusion image |
|-------|---------|--------------|
| Mini. | -0.16 | -0.09 |
| Max. | 0.65 | 0.46 |
| Mean | 0.10 | 0.10 |
| SD | 0.03 | 0.03 |

Statistical results

**Figure 4 The observed reflectance values and estimated value for the blue band using ESTARFM: November 22, 2018.**


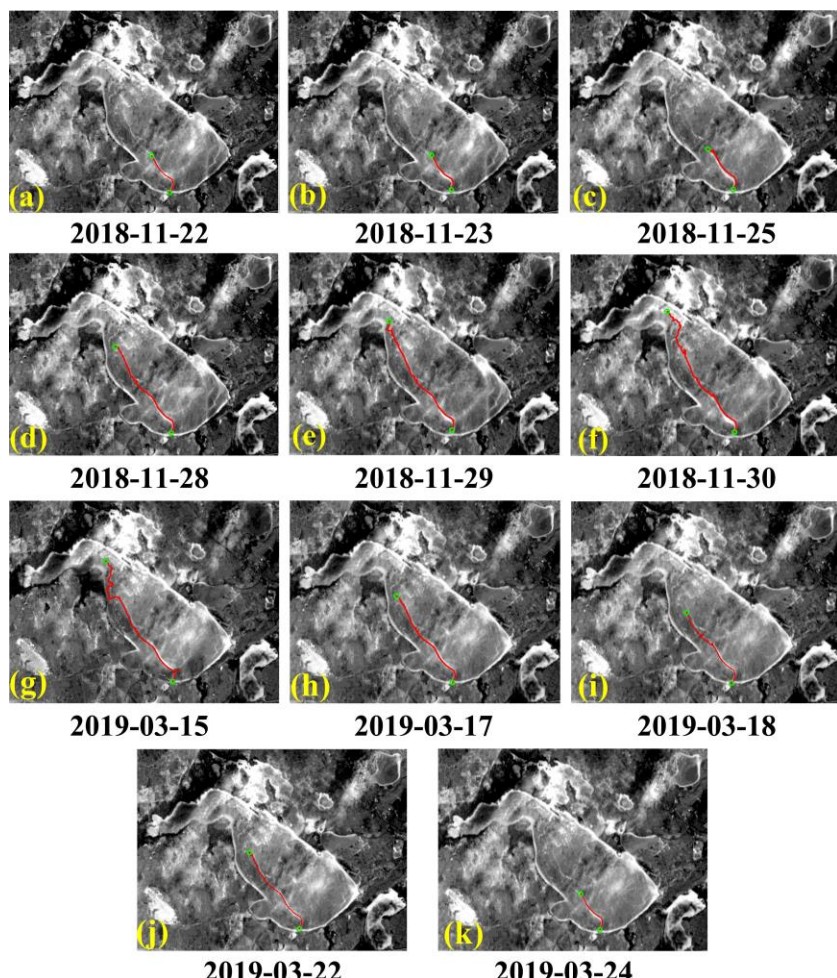

**Figure 5 The growth and decay process of ice ridges derived from the fusion data during the cold season from 2018 to 2019. Figure 5(a)-(f) displays the growth process from November 22, 2018, to November 30, 2018; Figure 5(g)-(k) displays the decay process lasted from March 15 to March 24, 2019.**



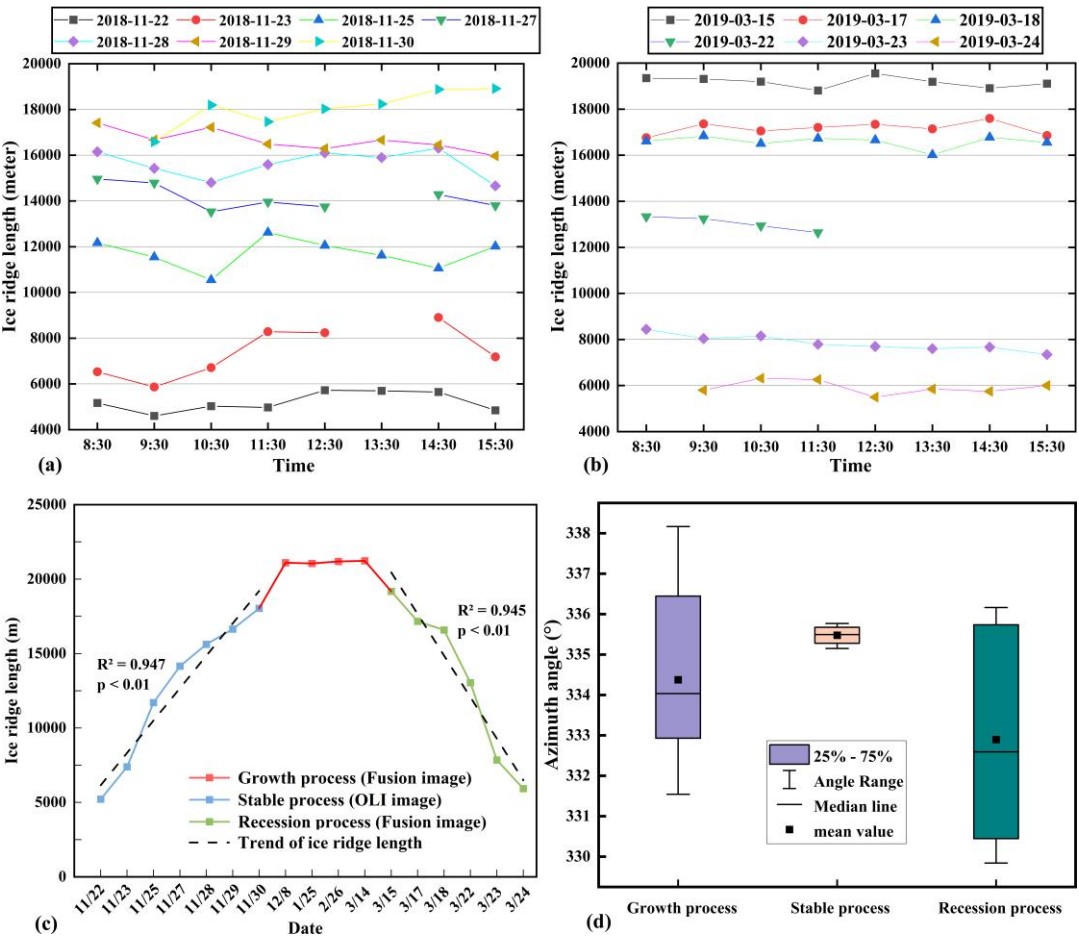

**Figure 6 The length changes of ice ridges from November 2018 to March 2019. The length and angle during the growth and recession process were extracted from Landsat-GOCI fusion data, and that of the stable process was extracted from Landsat data.**






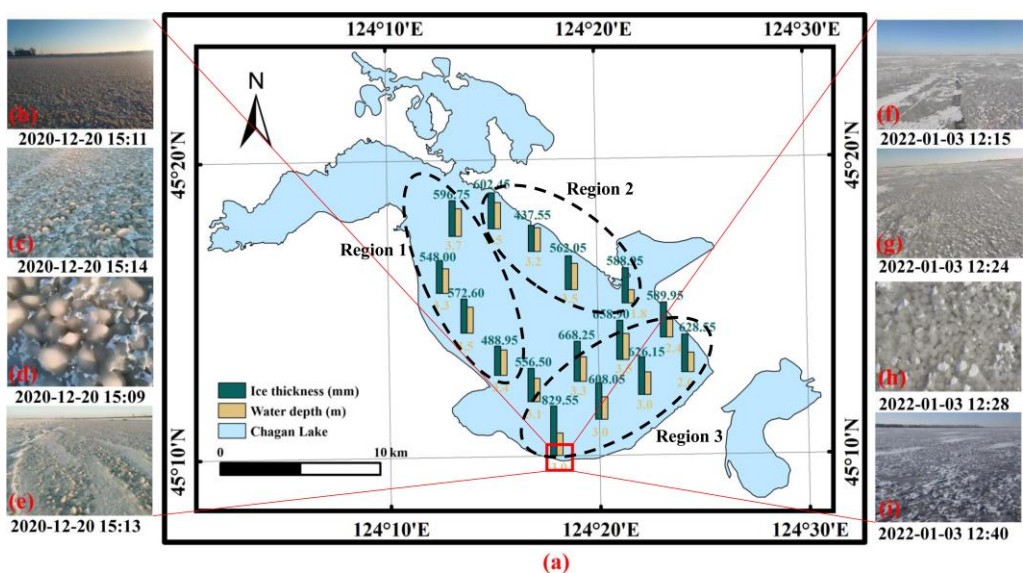

**Figure 7 The ice thickness of Chagan Lake was measured during the periods from January 2 to 4, 2022, and the field photograph.**

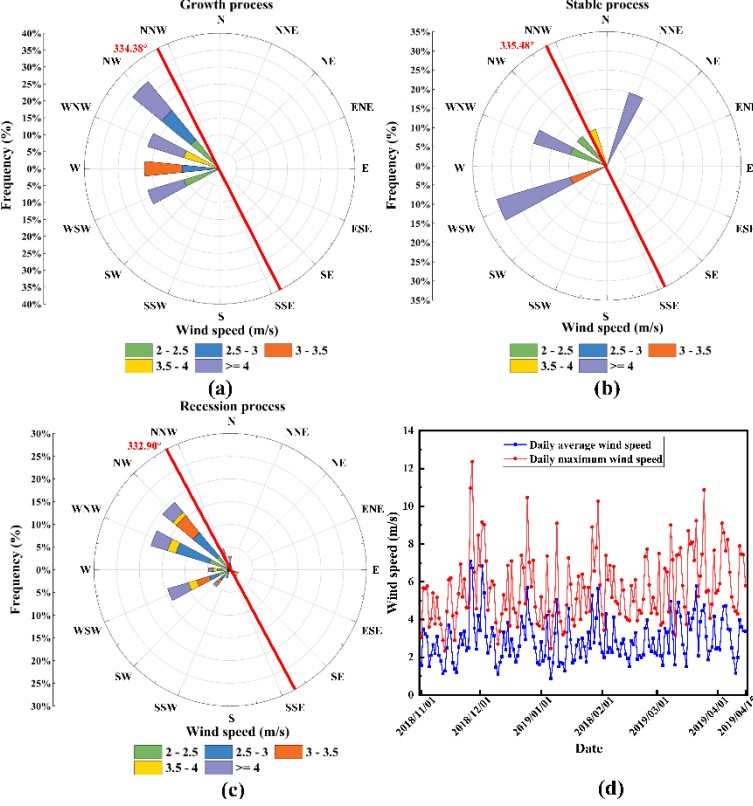

**Figure 8 The wind rose of Chagan Lake during the cold season of 2018: (a) growth process from November 22, 2018, to November 30, 2018; (b) stable process; (c) recession process from March 15 to March 24, 2019; (4) daily average and maximum weed speed from November 1, 2018, to April 15, 2019.**



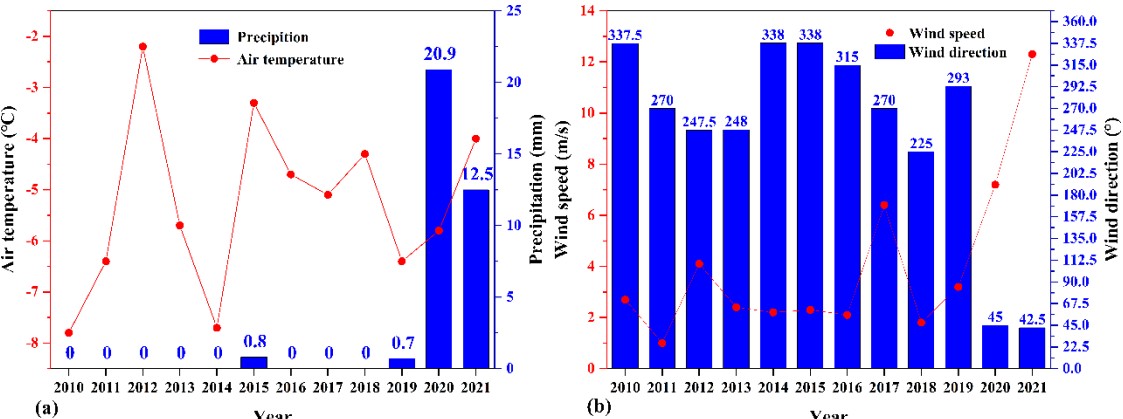

**Figure 9 The changes of climate changes around the freeze-up process from 2010 to 2022: (a) air temperature (°C) and precipitation (mm); (b) wind speed (m/s) and wind direction (°). The average values are calculated a week before and after the freeze-up date derived from MODIS.**
