# Peer review of "Fusion of Landsat 8 OLI and COCI for hourly monitoring surface morphology of lake ice with high resolution in Chagan Lake of Northeast China"

_The Cryosphere, 2022_

## Referee Comment (RC1)

This research applied image fusion of Landsat OLI and GOCI images using ESTARFM, then extracted linear structure on the lake ice surface of the high spatial-temporal fusion images and monitored the changes in surface morphology in Chagan Lake. Through the correlation analysis with meteorological factors, it showed that the surface morphology is closed related to wind direction, snowfall, and air temperature. This paper provided a good idea for monitoring lake ice surface morphology. However, there are still some improvement should be made and some mistakes need to be corrected in the manuscript.

1、 Line 11-14, "The surface morphology of lake ice undergoes remarkable changes under the combined influence of thermal and mechanical forces, which has been rarely observed by remote sensing. A large-scale linear structure has repeatedly appeared on satellite images of Chagan Lake in recent years." The first sentence said the surface morphology has been rarely observed by remote sensing, the second sentence said that a large-scale linear structure has repeatedly appeared on satellite images of Chagan Lake in recent years. So, why it was rarely observed before and appeared recently? Is it because of the remote sensing image with course resolution or there are few surface morphology before?

2、 Line14, "We prosed a method to extract linear structure on the lake ice surface." According to the manuscript, the extraction method of linear structure on the lake ice surface is not proposed in this paper, but your work before.

3、 Line 51-54, these sentences do not fit in here. You said, "multi-sensor, multi-temporal and multi-spatial resolution remote sensing image data have been successfully applied to monitoring the lake ice" in end of last paragraph. Then the common fusion methods should be described. While the high spatial-temporal resolution fusion such as Landsat and MODIS fusion is just one of these fusion methods.

4、 Line 134, Why are the reflectance images of Landsat band 2 and GOCI band 3 chosen to be fused? Why not the Pan band of OLI with 15m resolution?

5、 Line 138, "estarfm" should be capitalized.

6、 There are some problems in formula 1

(1) There are two "n" in the formula, k=m,n and i=1 to n

(2) L is "Landsat OLI", but not "MODIS "

(3) How to calculate the *Wi* and the *Vi*, and what is the convert coeffects?

(4) I think the formula is STARFM but not the ESTARFM

7、 Line 172, "The high $R^2$ between actual and predicted images was 0.935 on November 28, 2018, which proved that the fusion images are consistent with the remote sensing data." I think there is no Landsat OLI image on November 28, 2018, how can you do that?

8、 Line 176, "Estarfm" shoud be capitalized.

9、 Line 191, figure9 should be figure6.

10、 Line 214, "However, the ice ball was frozen beneath the ice surface, and the surface was relatively smooth." Is it said the ice ball of 2022? It should be clear in the manuscript.

11、 Line 257, same as the second comment.

12、 Line 275, "From Figure 9, the occurrence of ice ball needs to meet the strict requirements of climate conditions together during the frozen process:" It seems that the conditions for the occurrence of ice ball are summarized in Figure 9. However, line 281 is "During the past decade, only the cold season of 2020 and 2021 could meet all three conditions". Here , the logic is not clear.

---

## Author Comment (AC1)

The authors present a study on the morphological properties of ice cover on ~200 km2 large shallow Lake Chagan in China. The ice cover morphology was investigated in terms of formation of ice ridges, their evolution during the ice-covered period, and relationship to the wind force/direction. The central point of the ms is the exploration of the ability of satellite-based remote sensing with regard to quantification of the ice ridge properties. A second "parallel" story develops throughout the ms, discussing a distinct and rarely reported type of lake ice---the "ice balls"---which were encountered by the authors during their field campaigns in support of the remote sensing data analysis.

The subject of the study is suitable for "The Cryosphere" and can potentially be of interest for the journal's wide audience: seasonal lake ice is a relatively poorly investigated part of the cryosphere, which attracts growing attention of researchers during the last several decades. Indeed, ice cover on lakes with large (compared to the length scales based on the ice thermal expansion coefficient or on the length scales of the mechanical deformation) spatial dimensions tends to have a complex morphological structure with long-lasting ridges, cracks, stamukhi, and other quasi-regular features requiring a deeper investigation for correct understanding of the role played by seasonal ice cover in large lake dynamics and land-atmosphere interaction. In this sense, the authors present a valuable dataset and a well-supported methodology with a potential for expansion on other large lakes worldwide.

**Reply to comment:** Thank you for your comments concerning our manuscript entitled "The Capability of high spatial-temporal remote sensing imagery for monitoring surface morphology of lake ice in Chagan Lake of Northeast China" (tc-2022-175). Those comments are all valuable and very helpful for revising and improving our paper, as well as the important guiding significance to our work. We have carefully gone through the comments and made corrections accordingly, marked as red in the manuscript. Regarding English usage and grammar, we used a professional English editing service by Essentialink Language Service to improve our manuscript, and the certificate is provided as an attachment.

Essentialslink Language Services

**ENGLISH EDITING CERTIFICATE**

This document certifies that the manuscript listed below was edited for proper English language, grammar, punctuation, spelling, and overall style by one or more of the highly qualified native English-speaking editors at Essentialslink Language Services

**Manuscript title**

Fusion of Landsat 8 OLI and COCI for hourly monitoring surface morphology of lake ice with high resolution in Chagan Lake of Northeast China

**Authors**

Qian Yang, Xiaoguang Shi, Weibang Li, Kaishan Song, Zhijun Li, Xiaohua Hao, Fei Xie, Nan Lin, Zhidan Wen, Chong Fang and Ge Liu.

**Order No**

ELLSNOnt20221227174024296

**Date Issued**

December 30, 2022

[Figure]

This document certifies that the manuscript listed above was edited for proper English language, grammar, punctuation, spelling, and overall style. Neither the research content nor the author's intentions were altered in any way during the editing process. Documents receiving this certification should be Englishready for publication; however, the author has the ability to accept or reject our suggestions and changes. If you have any questions or concerns about this document or certification, please contact business@essentialslink.cn.

Essentialslink Language Services is a service of Qingdao Essentialslink Co., Ltd, which is committed to providing high quality services for researchers. To find out more about Essentialslink, visit https://www.essentialslink.cn

I had several major concerns raised when reading the manuscript:

- The story about the ball-shaped ice structures discovered by the authors on the lake surface is not connected to the declared subject of the study on the capability of satellite imagery for monitoring ice morphology. The phenomenon of "ice balls" per se is interesting for understanding the physics behind the processes of ice formation at different weather conditions, and the authors presented a reliable hypothesis on their formation supported by meteorological observations. However, it should be considered in a more consequent way and presented as a separate study in glaciological literature. Otherwise, the results will remain hidden under a wrong title and will only disturb the presentation of the actual topic of the study. As a separate study, the presentation of the "ball ice" should be accompanied by an extended discussion on frequency of its formation in waters of different types and geographical location and on its potential effects on the ice properties and under-ice conditions with analysis of information from other reports on the phenomenon. In addition to the works cited by the authors, the phenomenon was described in the literature on Lake Baikal under the russian term "kolobovnik", see, e.g.,

Granin, N.G., Aslamov, I.A., Kozlov, V.V. et al. (2019) Methane hydrate emergence from Lake Baikal: direct observations, modelling, and hydrate footprints in seasonal ice cover. Sci Rep 9, 19361. https://doi.org/10.1038/s41598-019-55758-8

Vologina, E. G., Granin, N. G., Vorobeva, S. S. et al. (2005). Ice-rafting of sand-silt material in South Baikal. Russian Geology and Geophysics, 46(4), 420-427.

and citations therein.

**Reply to comment:** We appreciate your recognition of our work, and we deleted the related contents with the ice ball. We presented the preliminary analysis of ice balls of Lake Sihai and Lake Chagan (Xie et al., 2020) [1], and we hope to map the spatial distribution of the ice balls, and explain the formation by local weather, the flow field, and the topography underwater. We are planning to write a paper focusing on ice balls in an extended English version.

Reference:

[1] Xie, F., Lu, P., Cheng, B., Yang, Q., and Li, Z.: Magical spherical ice (ice balls, ice eggs), Journal of lake sciences, 695-698, 2022(in Chinese).

- The discussion on the main topic of the study, the detection of ice ridges from satellite imaging, is rather short and superficial. To put the results in the right context, it should be extended with information on the potential application of the results in lake ice studies and comparative analysis against other publications on the same subject.

**Reply to comment:** Thank you for the professional suggestion, and we re-wrote this part (Line 255-294).

[revised manuscript text omitted]

- It is annoying to put language issues on the list of concerns. However, in this case, the authors have to perform hard and responsible work to make this text understandable to the reader. The text is full with repeated words, unfinished phrases, and sentences. Figures lack comprehensive legends and are overloaded with irrelevant information. Apart from a careful proofread, the help of a native speaker is recommended. Some of my remarks are provided in the (non-exhaustive) list of detailed comments below.

**Reply to comment:** Regarding English usage and grammar, we used a professional English editing service by Essentialink Language Service to improve our manuscript, and the certificate is provided as an attachment.

Line 14: "prosed" -> proposed

**Reply to comment:** Thank you for the careful check. We updated the Abstract, and delete the sentence. (Line 14).

Line 20: "closed related" -> closely related

**Reply to comment:** Thank you for the careful check, and we modified it as you suggest (Line 21).

Line 30: "scarce work studies" -> studies on… are scarce

**Reply to comment:** Thank you for the careful check, and we modified it as you suggest (Line 32).

Line 34: "has the advantages of" -> is

**Reply to comment:** Thank you for the careful check, and we modified it as you suggest (Line 36).

Line 44: "the cost is too expensive" -> costs are high

**Reply to comment:** Thank you for the careful check, and we updated the sentence as follows (Line 45-47).

*Although the temporal resolution of active microwave remote sensing data has been improved from 30 days (ERS) to daily return visits (Radasat-2), the optimized technique is too costly and more suitable for case studies of small or medium lakes.*

Line 45: "time series" -> "temporal coverage"

**Reply to comment:** Thank you for the careful check, and we modified it as you suggest (Line 47).

Line 61: "coarse and fine resolution and coarse resolution…" -> ?

**Reply to comment:** Thank you for the careful check, and we have made changes in the text (Line 63).

Lines 74-76: The abbreviations SAR and UAV are in the meantime widespread. Nevertheless, they should be better expanded.

**Reply to comment:** Thank you for the careful check, and we modified it as you suggest (Line 75-76).

Line 90: how the lake length of 107 km was determined? With the surface area of 253 km2, it would mean the lake "width" of less than 3 km. The map in Fig. 1 looks however different from that.

**Reply to comment:** Thank you for your careful question. The length of the lake mentioned in the text refers to the perimeter of the lake. The perimeter of the lake is

obtained by calculating the perimeter of the vector data of Lake Chagan without the recalculated area in Figure 1. We recalculated the length and area of Lake Chagan and updated them in the paper, and the perimeter of Lake Chagan are 329.72 km$^2$ and 201.03 km, respectively (Line 91-92). We used the geometric calculation of ArcGIS based on the vector of Lake Chagan.

[Figure]

Figure 1 The spatial distribution of Chagan Lake

Line 129: What is meant under "A 16-degree angle with an interval of 25°"? Reformulate in a clear way.

**Reply to comment:** Thank you for the careful check and we are sorry for the mistake. Sixteen directions describe the wind field in the winter, and the angle of wind direction in each direction is 22.5°. We update the sentence as follows (Line 135-138):

*Sixteen directions with an interval of 22.5 ° describe the wind direction, covering north (N), north northeast (NNE), northeast (NE), east northeast (ENE), east (E), east southeast (ESE), southeast (SE), south southeast (SSE), south (S), south southwest (SSW), southwest (SW), west southwest (WSW), west (W), west northwest (WNW), northwest (NW), north northwest (NNW).*

Line 151: "The morphological extraction": replace the section title with a meaningful one

**Reply to comment:** Thank you for the helpful suggestion, and we replace it with "The quantitative analysis of linear structures" (Line 164).

Line 153: "Canny operator": replace with "the Canny edge detection algorithm" and

provide a reference.

**Reply to comment:** Thank you for the professional suggestion. We updated it and add a new reference.

**Reply to comment:** Thank you for the careful check, and we have modified the whole paragraph.

*At the beginning, the Landsat-GOCI fusion images were transformed into binary images. We extracted the original linear network by the Canny edge detection algorithm (Canny, 1986) and then conducted edge detection to remove the outer boundaries. The morphological processing, including opening, filling, and eroding sequentially, was implemented for the internal part of the linear network. Then, the linear structure was derived from the largest connected domain of linear network without boundaries, and the length is calculated by the shortest path of largest connected domain. We connected the northmost and southmost ends into a straight line. The angle followed the definition of wind direction above. We compared auto-extraction and visual interpretation in our previous work (Hao et al., 2021). The $R^2$ values of the length and angle of 0.96 and 0.98, respectively, which proved the well performance of auto-extraction.*

**Line 155: "inter" -> inner (?)**

**Reply to comment:** Thank you for the careful check, and we modified it as you suggest (Line 168).

Lines 160-180: the whole paragraph is barely understandable and controversial. At Lines 160-170 the validation of a predicted image on Nov 22 is discussed and the correlation value of 0.93 is declared. On Lines 171-180 the correlation of 0.935 is

referred to the date of Nov 28 (???) What is the difference between the two validations? Why was Nov 28 additionally used to Nov 22? What kind of new information is provided at Lines 171-174 compared to the Lines 160-170? The paragraph has to be deeply revised, repeated information removed, and the results delivered in an unambiguous way.

**Reply to comment:** thank you for the professional questions. We re-wrote this part. We only carried out one validation on November 22, 2018, and November 28, 2018 is the wrong date. We update this paragraph as follows (Line 175-188):

*We predicted the fine images from two pairs of fine Landsat and coarse GOCI data to fill the data gap caused by the low revisit frequency of the Landsat. The two known pairs of data in the freeze-up process were captured on November 6, 2018, and December 8, 2018, and 53 fine ESTARFM fusion images were predicted from coarse GOCI images. The two known pairs of data of the break-up process were captured on February 26, 2019, and April 15, 2019, and 43 fine ESTARFM fusion images were predicted. Figure 3 compares the spatial distribution of the original images and predicted images on November 22, 2018. In the predicted images, the texture of the ground objects was maintained, and enlargement figures in Figure 3 (c) and Figure 3 (d) clearly display the distribution of linear structure. The predicted images were well consistent with the original images, and indicates a good fusion effect of ESTARFM. Figure 4 illustrates the scatter plots of the actual and predicted reflectance values along the 1:1 line. The $R^2$ value is 0.935, indicating that the predicted image was highly correlated with the actual image. The ranges of predicated and actual images were consistent; their mean reflectance values were both $0.10 \pm 0.03$. The performance of the ESTARFM results was limited by (1) the limited image pairs available during the cold season from 2018 to 2019; (2) the time lag between the predicted and actual images; (3) the inconsistency of capture time between the predicted images and two pairs of input images (Lu et al., 2019; Liu et al., 2018a). Therefore, the ESTARFM fusion images had a good performance and can provide reliable materials for further exploration.*

[Figure]

*Figure 3 The actual image observed on November 22, 2018 (a) and its prediction images by the ESTARFM (b). The lower row images (c) and (d) display the enlargement figure of red rectangle in upper row images (a) and (b).*

[Figure]

*Figure 4 Scatter plot of the real and the predicted reflectance by the ESTARFM for the blue band. The capture date was November 22, 2018.*

Line 183: "display" -> "displays"

**Reply to comment:** Thank you for the careful check, and we modified it as you suggest (Line 193).

Line 187:"liner" -> "linear"

**Reply to comment:** Thank you for the careful check, and we modified it as you suggest (Line 197).

Line 191: "Figure 9" -> Figure 6

**Reply to comment:** Thank you for the careful check. We deleted the related contents of ice ball, and Figure 9 is deleted (Line 198).

Line 200: remove "rapid"

**Reply to comment:** Thank you for the careful check, and we modified it as you suggest (Line 209).

Line 212: "ball" -> balls

**Reply to comment:** Thank you for the careful check, We deleted the related contents of ice ball.

Lines 213-214 -> insert "In 2022" (?) Otherwise, the sentence is senseless

**Reply to comment:** Thank you for the careful check. We deleted the related contents of ice ball.

Lines 218-224: The whole passage is completely distracting. "The ice thickness had the smallest value" - WHERE? The difference between summer 2021 and winter 2021 - the difference of WHAT? "Differences were not significant enough to explain what we observed.." WHAT did you observe? "The ice thickness… showed spatial coherence... especially with the ice thickness" - the phrase is senseless. The passage looks like a piece of a draft text understandable to the author only and inserted into the ms without any editing. As I mentioned above, the information on the "ice balls", as presented here, is irrelevant to the main subject of the study and should be completely removed for consistency. However, this presentation style is inacceptable for a scientific work and should be reconsidered by the authors before submitting it somewhere else.

**Reply to comment:** Thank you for the careful check, and we have modified the whole paragraph as follows.

*Considering the safety of traveling on ice, we conducted two field investigations during*

*the two recent cold seasons, from December 30 to 31, 2020, and January 2 to 4, 2022, respectively. We located ten sampling points along the linear structure on the satellite images and collected field photos of ice ridges and fractures (Figure 1). We further verified the large-scale fractures on the images as ice ridges. We divided the whole lake into three regions according to the surface morphology. Region 1 was distributed along the ice ridges. The surface of lake ice is uneven, ice fractures and ice ridges were widely distributed. Region 2 was distributed along the northeastern coast, where Chagan Lake Ice and Snow Fishing and Hunting Cultural Tourism Festival has been held at the end of December each year. Region 3 covered the southern part of Chagan Lake. The lake ice in Regions 2 and Region 3 were flat and smooth.*

*We also measured the ice thicknesses and water depths of 16 sampling points (Figure 7). The ice thicknesses in the winter of 2021 ranged from 437.55 mm to 668.25 mm, with an average value of 582.24 ± 58.14 mm. The average ice thicknesses of Regions 1, 2, and 3 were 551.58, 547.75, and 645.74 mm, respectively. The average water depths of Regions 1, 2, and 3 were 3.48, 2.99 and 3.00 m, respectively. Among the three regions, Region 2 had the smallest average values of ice thickness and water depth. The differences in water depth between the fall of 2021 and the winter of 2021 had an average value of 0.12 ± 0.05 m and a maximum value of 0.2 m. The water depth in winter was lower than that in fall, and the decreasing water level also was a cause of lake ice fracturing in winter (Leppäranta, 2015). The ice features first formed in the nearshore area of the southeast coast, where the water depth was relatively smaller than that in other regions in Figure 7. The ice thicknesses and water depths showed spatial coherence with the surface morphology.*

[Figure]

*Figure 7 The ice thickness (mm) and water depth(meter) of Chagan Lake was measured during the periods from January 2 to 4, 2022. Ice balls were discovered along the southern coast, and the location is marked with red square.*

Line 226: what is "wind rise" in this context? Revise phrasing

**Reply to comment:** Sorry for the spelling mistake. It should be wind rose, and we modified it (Line 236).

Line 227: how the freeze-up and break-up dates were defined?

**Reply to comment:** Thank you for your careful question. The lake ice phenology of cold season from 2018 to 2019 was extracted from the combined time series of the surface temperature of lake water provided by MOD11A1 and MYD11A1 product (Song et al., 2016; Hao et al., 2021). The freeze-up date is defined as the first day when the surface temperature of lake water is below 0℃ in winter; the break-up date is defined as the first day when the surface temperature of lake water is above 0℃ in spring (Line 130-134). We haven't published the related work yet.

Line 246: "uncanny" -> replace with a stylistically neutral word.

**Reply to comment:** Thank you for the careful check, and we modified it as you suggest (Line 248-252).

Line 247: "ball" -> balls

**Reply to comment:** Thank you for the careful check. We deleted the related contents of ice ball.

Lines 274-275: there are no reports from Finland in the cited works. What is "...and so on"? The majority of the reports on "ice balls" comes from the coastal ocean and Laurentian Great Lakes.

**Reply to comment:** Thank you for pointing this out, and we delete the description of ice ball in the new version.

Line 277: How was the exact threshold of -10°C determined? Why is it not -8°C or -12°C?

**Reply to comment:** Thank you for the helpful suggestion. and we delete the description of ice ball in the new version.

Line 480, Fig. 6: The legend lacks explanation of the panels a-d. Information on Panels a-b is barely understandable and has no reference to other Panels. The in-figure legend on Panel c seems to be wrong: red line refers to "stable process" not to "growth process". It is unclear what kind of statistics (spatial or temporal) is used in the box-whiskers chart in Panel d.

**Reply to comment:** Thank you for the useful suggestion, and we modified figure 6 and its title.

[Figure]

Figure 6 The changes of ice ridges during the cold season of 2018-2019: (a) length changes during the growth process from November 20 to November 30, 2018 measured from 53 ESTARFM-fused images; (b) length changes during the recession process from March 15 to 24, 2019, measured from 43 ESTARFM-fused images; (c) the daily average length; (d)the angles of ice ridges in different stages.

All figures are overloaded with unnecessary and unexplained information. They should be deeply revised to provide essential information in an unambiguous way.

**Reply to comment:** Thank you for the useful suggestion, and we modified all the figures and delete the unnecessary information. You can check it in Lines 480-505.

[Figure]

**Figure 1 The spatial distribution of Chagan Lake and field photographs. Figure 1 (a) is provided Landsat 8 OLI on February 10, 2019 with the band composite: R(5) + G(4) + B(3). Figure 1 (b)- (h) displays the field photographs captured in field investigations.**

[Figure]

**Figure 2 The workflow of this study.**

[Figure]

**Figure 3 The actual image observed on November 22, 2018 (a) and its prediction images by the ESTARFM (b). The lower row images (c) and (d) display the enlargement figure of red rectangle in upper row images (a) and (b).**

[Figure]

**Figure 4 Scatter plot of the real and the predicted reflectance by the ESTARFM for the blue band. The capture date was November 22, 2018.**

[Figure]

**Figure 5 The temporal changes of linear structure on fusion images of Lake Chagan during the cold season of 2018-2019.**

[Figure]

**Figure 6 The changes of ice ridges during the cold season of 2018-2019: (a) length changes during the growth process from November 20 to November 30, 2018 measured from 53 ESTARFM fusion images; (b) length changes during the recession process from March 15 to 24, 2019, measured from 43 ESTARFM fusion images; (c) the daily average length; (d)the angles of ice ridges in different stages.**

[Figure]

**Figure 7 The ice thickness (mm) and water depth (m) of Chagan Lake was measured during the periods from January 2 to 4, 2022.**

[Figure]

**Figure 8 The wind rose of Chagan Lake during the cold season of 2018: (a) growth process from November 22, 2018, to November 30, 2018; (b) stable process; (c) recession process from March 15 to March 24, 2019; (4) daily average and maximum weed speed from November 1, 2018, to April 15, 2019.**

---

## Author Comment (AC2)

This research applied image fusion of Landsat OLI and GOCI images using ESTARFM, then extracted linear structure on the lake ice surface of the high spatial temporal fusion images and monitored the changes in surface morphology in Chagan Lake. Through the correlation analysis with meteorological factors, it showed that the surface morphology is closed related to wind direction, snowfall, and air temperature. This paper provided a good idea for monitoring lake ice surface morphology. However, there are still some improvement should be made and some mistakes need to be corrected in the manuscript.

**Reply to comment:** Thank you for your comments concerning our manuscript entitled "The Capability of high spatial-temporal remote sensing imagery for monitoring surface morphology of lake ice in Chagan Lake of Northeast China" (tc-2022-175). Those comments are all valuable and very helpful for revising and improving our paper, as well as the important guiding significance to our work. We have carefully gone through the comments and made corrections accordingly, marked as red in the manuscript.

1、 Line 11-14, "The surface morphology of lake ice undergoes remarkable changes under the combined influence of thermal and mechanical forces, which has been rarely observed by remote sensing. A large-scale linear structure has repeatedly appeared on satellite images of Chagan Lake in recent years." The first sentence said the surface morphology has been rarely observed by remote sensing, the second sentence said that a large-scale linear structure has repeatedly appeared on satellite images of Chagan Lake in recent years. So, why it was rarely observed before and appeared recently? Is it because of the remote sensing image with course resolution or there are few surface morphology before?

**Reply to comment:** Thank you for your question, the sentence is not appropriate herein, and we modified it, as follows (Line 12-13).

*The surface morphology of lake ice remarkably changes under the combined influence of thermal and mechanical forces. However, research on the surface morphology of lake ice and its interaction with climate is limited.*

Then we come to your question. A large-scale linear structure has repeatedly appeared

on satellite images of Chagan Lake in recent years. We checked the quick images of three sensors of Landsat since 1986, including TM, ETM+, and OLI (Figure 1). A similar linear structure had been found on the Landsat quick images during 18 of those 35 cold seasons.

[Figure]

Figure 1. The quick images of Landsat with similar cracks. We checked three sensors of Landsat for 35 cold seasons since 1986, including TM, ETM+, and OLI.

So, why it was rarely observed before and appeared recently? The lake area of Chagan is only 372 km$^2$, and the small lake hasn't received wide attention. Scarce work has been carried out for frozen lakes in Northeast China. Moreover, the image quality limited the remote sensing monitoring. Figure 2 presents the satellite images of Landsat OLI 8 with the best quality for each year. The existence of snow cover and cloud cover makes it difficult to extract the length and angle of linear structure. Only the images during 2018-2019 provide suitable materials for further exploration.

[Figure]

Figure 2. The satellite images of the frozen Chagan Lake with the best quality for the cold seasons from 2013 to 2020.

2、 Line14,"We prosed a method to extract linear structure on the lake ice surface."

According to the manuscript, the extraction method of linear structure on the lake ice surface is not proposed in this paper, but your work before.

**Reply to comment:** Thank you for your careful question. We re-write this sentence in the abstract as follows (Line 16-18).

*We merged the Landsat and GOCI images using an enhanced spatial and temporal adaptive reflectance fusion model (ESTARFM), and extracted the lengths and angles of the linear structure. We monitored the hourly changes in the surface morphology during the cold season from 2018 to 2019.*

3、 Line 51-54, these sentences do not fit in here. You said, "multi-sensor, multitemporal and multi-spatial resolution remote sensing image data have been successfully applied to monitoring the lake ice" in end of last paragraph. Then the

common fusion methods should be described. While the high spatial-temporal resolution fusion such as Landsat and MODIS fusion is just one of these fusion methods.

**Reply to comment:** Thank you for the professional question. The logic is not reasonable herein, and we modified as below (Line 50-51):

*Although multi-source remote sensing is available to monitor lake ice processes, single-sensor remote sensing data cannot simultaneously achieve accurate remote sensing monitoring and high frequency.*

4、 Line 134, Why are the reflectance images of Landsat band 2 and GOCI band 3 chosen to be fused? Why not the Pan band of OLI with 15m resolution?

**Reply to comment:** This is a good question and giver us insight into our work. we chose the two bands considering the spectral feature of the water body. Table 1 compares the band information of Landsat and GOCI. From it, we can find that the band range of band 3 of GOCI (480-500 nm) overlaps with that of band 2 of Landsat 8 OLI (450-515 nm). The water body had relatively strong reflectance in the blue band (400-480 nm), and the blue band images clearly display the linear structure. The table and the corresponding content have been added to the new version of the manuscript.

Table 1 The comparison of band information between GOCI and Landsat. The selected bands for merging Landsat and GOCI are marked with red.

| Band | GOCI | | Landsat 8 OLI | |
|---|---|---|---|---|
| | Band centre (nm) | Bandwidth (nm) | Band centre (nm) | Bandwidth (nm) |
| Band 1 | 402-422 | 20 | 433-453 | 20 |
| Band 2 | 433-453 | 20 | 450-515 | 65 |
| Band 3 | 480-500 | 20 | 525-600 | 75 |
| Band 4 | 545-565 | 20 | 630-680 | 50 |
| Band 5 | 650-670 | 20 | 845-885 | 40 |
| Band 6 | 675-685 | 10 | 1560-1660 | 100 |
| Band 7 | 735-755 | 20 | 2100-2300 | 200 |
| Band 8 | 845-885 | 40 | 500-680 | 180 |

5、 Line 138, "estarfm" should be capitalized.

**Reply to comment:** Thank you for your suggestion, we have modified the corresponding content in the text (Line 148).

6、 There are some problems in formula 1

(1) There are two "n" in the formula, k=m,n and i=1 to n

(2) L is "Landsat OLI", but not "MODIS "

(3) How to calculate the *Wi* and the *Vi*, and what is the convert coeffects?

(4) I think the formula is STARFM but not the ESTARFM

Response: Thank you for your professional question. After our confirmation that Equation 1 is the formula of STARFM model, we have modified it to the formula of ESTARFM model in the text with detailed explanation of the formula.

**Reply to comment:** We feel very sorry for the errors, and we updated the expressions in the new version of the manuscript (Line 149-163).

*The ESTARFM, in which two pairs of Landsat and GOCI images were used to generate spatial-temporal fusion data, was proposed by Zhu et al. (2010) based on the STARFM. Firstly, the coarse GOCI data were projected and resampled to a fine Landsat image at two known times tm and tn. Secondly, similar neighborhood pixels were searched with a moving window by setting spectral differences. Thirdly, we calculated the normalized weight of each similar pixel by considering the spatial, spectral, and temporal differences. Then, the coarse GOCI values were transferred to fine Landsat data using the pixel-based conversion coefficients in the linear regression. Finally, the coarse GOCI data at the same time were used to calculate the fine fusion data at the predicted time (tp), expressed as follows (Liu et al., 2021; Bai et al., 2017; Zhu et al., 2010):*

$$L_b\left(x_{w/2}, y_{w/2}, t_p\right) = T_m \times L_{bm}\left(x_{w/2}, y_{w/2}, t_p\right) + T_n \times L_{bn}\left(x_{w/2}, y_{w/2}, t_p\right) \qquad (1)$$

*where $L_b$ ($x_{w/2}$, $y_{w/2}$, $t_p$) is the final predicted fine-resolution reflectance at the prediction time $t_p$; w represents the size of the moving window, and the corresponding center is ($x_{w/2}$, $y_{w/2}$); $L_{bk}$ ($x_{w/2}$, $y_{w/2}$, $t_p$) is the fine-resolution reflectance at $t_k$ (k = m or n) at the base date; Tk is the time weight, calculated from the magnitude of the detected change*

*in the reflectance of the coarse spatial resolution image between tm and tn and the*
*prediction moment tp;*

$$T_k = \frac{1/\left|\sum_{j=1}^{w}\sum_{i=1}^{w}C\left(x_j,y_i,t_k\right)-\sum_{j=1}^{w}\sum_{i=1}^{w}C\left(x_j,y_i,t_p\right)\right|}{\sum_{k=m,n}\left(1/\left|\sum_{j=1}^{w}\sum_{i=1}^{w}C\left(x_j,y_i,t_k\right)-\sum_{j=1}^{w}\sum_{i=1}^{w}C\left(x_j,y_i,t_p\right)\right|\right)},\left(k=m,n\right) \qquad (2)$$

*where C(xj, yi, tk) and C(xj, yi, tk) denote the image element values of similar image*
*elements (xi, yj) within the moving window of the coarse spatial resolution image at the*
*reference moment tk and prediction moment tp, respectively.*

7、 Line 172, "The high R2 between actual and predicted images was 0.935 on
November 28, 2018, which proved that the fusion images are consistent with the remote
sensing data." I think there is no Landsat OLI image on November 28, 2018, how can
you do that?

**Reply to comment:** We are sorry for the mistake and thank you for the mistake. The
correct date is November 22, 2018, and we have modified it (Line 183-184).

Line 176, "Estarfm" shoud be capitalized.

**Reply to comment:** Thank you for your suggestion, we have modified the
corresponding content in the text (Line 185).

Line 191, figure 9 should be figure6.

**Reply to comment:** Thank you for your careful check, we delete the Figure 9.(Line
198).

10、 Line 214, "However, the ice ball was frozen beneath the ice surface, and the
surface was relatively smooth." Is it said the ice ball of 2022? It should be clear in
the manuscript.

**Reply to comment:** Thank you for your suggestion, and we delete the related contents
of ice ball (Line 217-221).

11、 Line 257, same as the second comment.

**Reply to comment:** Thank you for the helpful suggestion, we updated as you suggest as below (Line 302-303).

*We calculated the lengths and the angles of the linear structure on the fusion images in the freeze-up and break-up processes during the cold season from 2018 to 2019.*

12、 Line 275, "From Figure 9, the occurrence of ice ball needs to meet the strict requirements of climate conditions together during the frozen process:" It seems that the conditions for the occurrence of ice ball are summarized in Figure 9. However, line 281 is "During the past decade, only the cold season of 2020 and 2021 could meet all three conditions". Here, the logic is not clear.

**Reply to comment:** Thank you for the professional suggestion. and we delete the related contents of the ice ball. This paragraph has been deleted。